# Determinants of preterm birth among women delivered in public hospitals of Western Ethiopia, 2020: Unmatched case-control study

**Muktar Abadiga**[1]*, **Bizuneh Wakuma**[1], **Adugna Oluma**[1], **Ginenus Fekadu**[2], **Nesru Hiko**[1], **Getu Mosisa**[1]

**1** School of Nursing and Midwifery, Institute of Health Sciences, Wollega University, Nekemte, Ethiopia,
**2** School of Pharmacy, Institute of Health Sciences, Wollega University, Nekemte, Ethiopia

* muktarabadiga@gmail.com

## Abstract

### Background

Worldwide, preterm birth accounts for 1 million deaths of infants each year and 60% of these deaths occur in developing countries. In addition to the significant health consequences on the infant, preterm birth can lead to economic costs. There was a lack of study in western Ethiopia, and most of those studies conducted in other parts of a country were based on card review with a cross-sectional study design. The risk factors of preterm birth may vary from region to region within the same country due to variation in socioeconomic status and health care service coverage. Therefore, this study aimed to identify determinants of preterm birth in western Ethiopia.

### Methods

An institutional-based case-control study was conducted from February 15 to April 15, 2020, in western Ethiopia. The eligible 188 cases and 377 controls were randomly selected for this study. Cases were women who gave birth after 28 weeks and before 37 completed weeks of gestation, and controls were women who gave birth at and after 37 weeks of gestation from the first day of the last normal menstrual period. Data were collected by a structured interviewer-administered questionnaire. The collected data were entered into Epi info version 7 and exported to SPSS version 21 for analysis. Multivariable logistic regression was used to identify determinants of preterm birth at P-value <0.05.

### Results

From a total of 565 eligible participants, 516 (172 cases and 344 controls) participated in this study with a response rate of 91.3%. The result of the multivariable analysis shows that mothers who developed pregnancy-induced hypertension (AOR = 3.13, 95% CI; 1.78, 5.50), only one time ANC visits (AOR = 5.99, 95% CI; 2.65, 13.53), experienced premature rupture of membrane (AOR = 3.57, 95% CI; 1.79, 7.13), birth interval less than two years (AOR = 2.96, 95% CI; 1.76, 4.98), developed anemia during the current pregnancy (AOR =

**Data Availability Statement:** All relevant data are within the manuscript and its Supporting Information files.

**Funding:** This study was funded by the Wollega university. The funders had no role in study design, data collection and analysis, decision to publish, or preparation of the manuscript.

**Competing interests:** The authors have declared that no competing interests exist.

**Abbreviations:** AIDS, Acquired immunodeficiency syndrome; ANC, Antenatal care; AOR, Adjusted odds ratio; APH, Antepartum hemorrhage; CI, Confidence interval; COR, Crude odd ratio; CRH, Corticotrophin-releasing hormone; DM, Diabetes mellitus; HIV, Human immunodeficiency virus; LMP, Last menstrual period; MUAC, Mid-upper arm circumference; PIH, Pregnancy-induced hypertension; PROM, Premature rupture of membrane; PTB, Preterm birth; SD, Standard deviation; SPSS, Statistical package for social science; VIF, Variance inflation factor.

4.20, 95% CI; 2.13, 8.28) and didn't get dietary supplementation during the current pregnancy (AOR = 2.43, 95% CI; 1.51, 3.91) had statistically significant association with experiencing preterm birth.

## Conclusion

Antenatal care service providers should focus on mothers with pregnancy-induced hypertension, premature rupture of membrane, and anemia during pregnancy, and refer to the senior experts for early management to reduce the risk of preterm delivery. Antenatal care services such as counseling the mother on the benefit of dietary supplementation during pregnancy, antenatal care follow up, and lengthening birth interval should be integrated into the existing health extension packages. New and inclusive strategies such as the establishment of comprehensive mobile clinic services should also be designed to reduce the burden of preterm birth among women living in the rural community. Lastly, we recommend future researchers to conduct longitudinal and community-based studies supplemented with qualitative methods.

## Introduction

Although under-five mortality rates were reducing over the past years, neonatal mortalities have been shown less improvement [1]. World Health Organization (WHO) estimates that about 8 million infants die each year worldwide, from which 1 million are due to preterm birth (PTB) [2]. In Ethiopia, it was estimated that about 320,000 births are PTB each year from the total newborns in the country [3]. PTB is one of the most common causes of neonatal death and is defined as a baby born too early or before 37 completed weeks of gestation from the first day of the last normal menstrual period [4]. PTB can be occurred spontaneously or is initiated by clinicians for different medical or nonmedical reasons before 37 weeks of gestation [5].

Globally, about 15 million infants were born prematurely, of which the highest proportion (60–80%) occurred in low and middle-income countries [6]. The WHO estimates that the prevalence of PTB ranges between 5–18% across 184 countries around the world [7]. The rate of PTB is about 12.8% in South Asia and 60% in sub-Saharan Africa, 10.2% in Central and Eastern European centers [8, 9], 15.8% in the French Caribbean population of African descent [10], and 13.7% in Saudi Arabia [11]. In African countries, the magnitude of PTB is 18.3% in Kenya [12], 9.26% in Algeria [13], 11.8% in Nigeria [14] and 16.3% in Malawi [15]. In Ethiopia, the rate of PTB is about 16.15% in the capital city Addis Ababa [16], 13.3% in Axum and Adwa [17], 12.8% in Debretabor town [18], and 16.9% in Shire Sihul hospital [19].

Worldwide, around 1 million children die each year due to complications related to PTB and 60% of these complications occur in developing countries [20]. Literature showed that PTB accounts for 28% of all 4 million annual early neonatal deaths [21]. In Ethiopia, PTB is the first cause of neonatal death accounting for 34% and contributes to 12.5% of deaths of under five children [22]. PTB is the second leading cause of death for children under-five years of age next to pneumonia, and about 24,400 less than 5 years of children die due to the direct effect of PTB in Ethiopia [23]. It remains a public health issue responsible for neonatal morbidity and mortality especially in low-income countries despite the improved antenatal coverage [24].

PTB has a greater risk of developmental disabilities and growth problems leading to 75% of perinatal deaths and 50% of neurological abnormalities [25]. It leads to short-term and long-

term problems in motor development, impairment in behavior, and poor academic performance in later life [26]. In addition to the significant health consequences to the infant, PTB can lead to economic costs for families, communities, and the nation at large [27]. It also has impacts on the health care system demanding rehabilitation service, special education placement, and specialized health care professionals [28]. More than 90% of preterm babies die in developing countries within the first few days of life while only less than 10% die in high-income countries [7].

Literature showed that extreme maternal age [29, 30], pregnancy induced hypertension (PIH) [11, 15, 31–36], antepartum hemorrhage (APH) [11, 33, 35], fewer antenatal care (ANC) visits [29, 33, 34, 36, 37], decreased hemoglobin level [17, 18, 31], multiple pregnancy [16, 34, 36], short birth interval [16, 29, 36], premature rupture of membrane (PROM) [11, 17, 33, 35, 36], maternal malnutrition [17, 38], chronic medical illness [16, 29] and being HIV sero-positive mothers [15, 17, 32] were some of the factors which contributed to the occurrence of PTB.

Although different interventions have been employed to prevent as well as to improve the outcome of PTB, the burden of PTB remains high in developing countries including Ethiopia, and control strategies have been given little attention [39]. Due to the enormous economic and emotional burden of PTB, identifying the risk factors for PTB has the potential to help in preventing the impacts. It is important to understand the risk factors of PTB especially in developing countries in which the rate of PTB is high. However, no study was conducted on the determinants of PTB in the western part of Ethiopia. Due to variations in socioeconomic status and health care service coverage, the risk factors of PTB may vary from region to region and time trends even within the same country. On the other hand, most of the studies conducted in other parts of a country were based on card review, and therefore, the risk factors of PTB were not fully addressed. On the contrary, this study was based on primary data and included many risk factors that would be associated with PTB. Besides, most of the studies conducted in Ethiopia were cross-sectional and this study used a case-control study design that is stronger than cross-sectional. Therefore, this study aimed to identify the determinants of PTB in western Ethiopia. The finding of this study is important for policymakers and health care workers by providing important information related to risk factors of PTB in designing an effective strategy to prevent and control PTB.

## Methods

### Study design, setting, and population

A hospital-based prospective unmatched case-control study was employed. This study was conducted in public hospitals of Wollega zones, western Ethiopia from February 15 to April 15, 2020. The total population of Wollega zone is about 3,345,675, from which 1,739,751 are females and 1,605,924 are males [40]. For administrative purposes, the Wollega zone is divided into 4 independent zones, namely; Horro Guduru Wollega zone, West Wollega zone, Kellem Wollega zone, and East Wollega zone. According to each zone's health bureau report, the Wollega zone has 13 primary hospitals, 9 general hospitals, 2 comprehensive specialized hospitals, 401 health posts, and 102 health centers. We conducted this study in six randomly selected hospitals found in Wollega zones, namely; Nedjo general hospital, Gimbi general hospital, Nekemte specialized hospital, Arjo hospital, Wollega university specialized hospital, and Shambu general hospital. According to a report from the zone, an estimated total number of 85,345 births have been registered annually in the zones. In this study, the source population was all mothers who gave birth in public hospitals of the Wollega zones and the study population was all women who gave birth at the randomly selected hospitals of Wollega zones during

the study period. All immediate postnatal women who gave birth at the selected hospitals of Wollega zones during the study period were included in the study. Women with unknown last menstrual period (LMP) or not reliable ultrasonography (not early taken at ≤20 completed weeks of gestation) and unable to communicate due to serious medical illness were excluded. Cases (preterm births) were women who gave birth after 28 weeks (fetal viability) and before 37 completed weeks of gestation from the first day of the last normal menstrual period. Controls were women who gave birth at and after 37 weeks of gestation from the first day of the last normal menstrual period. Post-term births occur after 37 weeks of gestation and are not similar to preterm births, and therefore it was included under controls in this study. On the other hand, deliberately or medically terminated pregnancies before 37 and after 28 weeks of gestation were also included as preterm birth.

## Sample size determination and sampling techniques

The Sample size was calculated using the double population proportion using EPI-Info 3.5.1 version statistical software. The proportion of experiencing PROM, preeclampsia/eclampsia, ANC <4 times, and APH in the current pregnancy was used to determine the sample size from a study done in Ghana [33]. Having experience of PROM was chosen as an independent variable since it brought a higher sample size among other computed explanatory variables. The assumptions for the sample size calculation were as follows: the proportion of mothers having experience of premature rupture of membrane among controls, a minimum detectable odds ratio of 2, confidence level of 95% (Zα/2 = 1.96), power of 80% (Zβ = 0.80) and a case to control ratio of 1:2 and proportion of case among an exposed group (premature rupture of membrane) of 35.4% [33]. After adding a 5% non-response rate, the total calculated sample size was 565 (188 cases and 377 controls).

First, six hospitals were randomly selected from the public hospitals that provide institutional delivery services in the Wollega zone. Then, the number of cases and controls were proportionally allocated to each hospital based on the number of mothers who gave birth at each selected hospital within four months before the data collection time. Then, the average number of mothers expected to give birth within 2 months in each of the respective hospitals was estimated. Finally, the eligible case was selected consecutively and the consecutive two controls were selected until the required sample size was achieved. Accordingly, we included 34 cases and 68 controls from Nekemte specialized Hospital, 32 cases and 64 controls from Wollega university specialized hospital, 20 cases and 40 controls from Arjo Hospital, 43 cases and 86 controls from Gimbi general hospital, 29 cases and 58 controls from Shambu Hospital and 30 cases and 60 controls from Nedjo hospital.

## Data collection tool and procedure

Data were collected by a structured interviewer-administered questionnaire adapted from the Ethiopian Demographic and Health Survey and other similar studies [23, 29, 33–35, 38, 41] and necessary modifications were done. The outcome variable was PTB and the exposure variables were socio-demographic variables, gynecologic and obstetric related factors, pre-conception related variables, nutritional and dietary related factors, behavior-related factors, mothers' history of pre-existing medical illness, and health facility-related factors (S1 Questionnaire). We used LMP date and ultrasonography finding (if performed at ≤20 completed weeks of gestation) to estimate gestational age (GA). If the LMP date and ultrasound date don't correlate/ disparity happened, defaulting to ultrasound for GA assessment is required and therefore we took the ultrasound date based on the American College of Obstetricians and Gynecologists (ACOG) recommendation [42]. Women with unknown LMP and the ultrasound

measurement not taken at an appropriate time (at ≤20 completed weeks of gestation) were excluded from the study. The interview was held in a separate room after a woman is stabilized and ready to be discharged. In addition to the interview, the data collectors abstracted clinical data by reviewing the mothers' and the babies' medical records. The mother's mid-upper arm circumference was measured using a flexible non-stretchable tape measure. Maternal hemoglobin level was reviewed from mothers' cards to determine anemia. The data was collected by 12 trained BSc midwives recruited from other hospitals not included in this study for 2 months. Six Master of Science qualified midwives supervised the overall data collection process.

## Data quality control

The questionnaire was translated to the local language Afan Oromo and then back to English by two different language experts to check for consistency. Five percent of the questionnaire (32 study participants) was pre-tested at the same study area 5 days before data collection and modification were made based on pre-test results. Two days training on the objectives of the study, sampling technique, ethical consideration, and data collection techniques were given for data collectors and supervisors. Continuous follow-up and supervision of data collection were made by the supervisors. The collected data were checked by the supervisor daily for completeness.

## Data processing and analysis

The collected data were coded, cleaned, and entered into Epi info version 7 and exported to SPSS version 21 for analysis. Descriptive statistics like frequencies and percentages were performed. Some categories of the variables with few numbers of participants in the cases and /or controls were merged to fulfill the assumptions of the binary logistic regression. Bivariable logistic regression analysis was used to see the unadjusted effect of each independent variable on the dependent variable and variables which have P-value of less than 0.25 were entered into a multivariable logistic regression model. Multivariable logistic regression was conducted to identify independent determinants of PTB. Model fitness was tested with the Hosmer-Lemeshow goodness of fit test and omnibus tests of model coefficients. Variance inflation factor (VIF) and tolerance tests were also used to check multicollinearity. The adjusted odds ratio (AOR) with a 95% confidence interval (CI) was calculated to determine the strength of an association. A P-value of $< 0.05$ was considered statistically significant in multivariable logistic regression.

## Ethics approval and consent to participate

The study was approved by the institutional review boards of Wollega University ethical review board with approval ID: HIS/213/20. A permission letter was obtained from each hospital administrative office. All participants of the study were provided written consent, clearly stating the objectives of the study and their right to refuse. Then, written informed consent was obtained from the study participants. For minors, informed consent was received from their parents or legal guardians. To ensure confidentiality, names, or identifying information was not indicated on the questionnaires. Mothers were interviewed in private rooms to ensure their privacy. The filled questionnaires were carefully handled ensuring confidentiality and were kept under the secured custody of the corresponding author.

## Results

### Sociodemographic characteristics of the study participants

From a total of 565 eligible participants (188 cases and 377 controls), 516 respondents (172 cases and 344 controls) were participated in the study making a response rate of 91.3%. The

age of mothers ranges from 15–48 years with the mean and standard deviation (±SD) of 28.71 and ±6.24 respectively. The majority of the study participants, 148 (86.0%) of the cases and 297 (86.3%) of the controls were Oromo in ethnicity. Regarding religion, 89 (51.7%) of the cases and 179 (52.0%) of the controls were protestant religion followers. Thirty-nine (22.7%) of the cases and 77 (22.4%) of the controls had no formal education. Regarding monthly income, 34 (19.8%) of the cases and 108 (31.4%) of the controls get a monthly income of 1000–2000 Ethiopian birr. Concerning residence, about 116 (67.4%) of the cases and 214 (62.2%) of the controls were rural dwellers (Table 1).

## Obstetrics related characteristics of the study participants

Sixty-five (37.8%) of the cases and 129 (37.5%) of the controls have greater than four births. Regarding the use of family planning, 128 (74.4%) of the cases and 253 (73.5%) of the controls had used family planning before the current pregnancy. About 121 (70.3%) of the cases and 278 (80.8%) of the controls were planned their pregnancy. Eighty (46.5%) of the cases and 183 (53.2%) of the controls were attended ANC more than or equal to four times. Regarding birth space, 118 (68.6%) of the cases and 288 (83.7%) of the controls were more than or equal to 2 years. Concerning the previous history of PTB, 156 (90.7%) of the cases and 316 (91.9%) of the controls had no history of PTB. Ninety-one (52.9%) of the cases and 208 (60.5%) of the controls were delivered through spontaneous vaginal delivery. One hundred forty (81.4%) of the cases and 321 (93.3%) of the controls didn't experience PROM, and 134 (77.9%) of the cases and 294 (85.5%) of the controls had no history of abortion (Table 2).

## Medical history related-characteristics of the study participants

The majority of the study participants, 294 (85.4%) of the controls and 137 (79.7%) of the cases had no history of hypertension. About 339 (98.5%) of the controls and 167 (97.1%) of the cases had no history of cardiac diseases. Concerning the history of diabetes mellitus (DM), 165 (95.9%) of the cases and 316 (91.9%) of the controls had no diabetes. Concerning HIV status, 156 (90.7%) of the cases, and 336 (97.7%) of the controls were HIV seronegative. The majority of the study participants, 136 (79.1%) of the cases and 318 (92.4%) of the controls have no anemia during the current pregnancy. About 165 (95.9%) of the cases and 328 (95.3%) of the controls have no malaria during pregnancy. The proportion of sexually transmitted disease and pregnancy-induced hypertension was slightly higher among cases 15 (8.7%) and 48 (27.9%) than controls 20 (5.8%) and 35 (10.2%) respectively (Table 3).

## Social and behavioral related characteristics of mothers

One hundred fifty-seven (91.3%) of the cases and 317 (92.2%) of the controls had no history of physical abuse. Regarding the use of traditional medicine, 162 (94.2%) of the cases and 325 (94.5%) of the controls were not used traditional medicine. About 90 (52.3%) of the cases and 226 (65.7%) of the controls had dietary supplementation during the current pregnancy. Regarding maternal social support, 111 (64.5%) of the cases and 225 (65.4%) of the controls have social support. The proportion of substance use among cases (20.9%) was approximately twice of substance use in controls 42 (12.2%). Similarly, the percentage of mothers who experienced stress was higher among cases (24.4%) than controls (19.2%).

## Determinants of preterm birth

Bivariable logistic regression analysis showed that age at marriage, ethnicity, age of mother, husband & mothers occupation, residence, monthly income, time to reach a health facility,

**Table 1. Sociodemographic characteristics of mothers attending birth at public hospitals of Western Ethiopia, 2020 (n = 516; cases: 172 and controls: 344).**

| Variables | Category | Cases N (%) | Controls N (%) | Total N (%) |
|---|---|---|---|---|
| **Age of mother** | 15–24 years | 50(29.1) | 87(25.3) | 137(26.6) |
| | 25–34 years | 83(48.3) | 192(55.8) | 275(53.3) |
| | ≥35 years | 39(22.7) | 65(18.9) | 104(20.2) |
| **Age at first marriage** | <18 years | 18(10.5) | 50(14.5) | 68(13.2) |
| | 18–23 years | 137(79.7) | 267(77.6) | 404(78.3) |
| | >23 years | 17(9.9) | 27(7.8) | 44(8.5) |
| **Ethnicity** | Oromo | 148(86.0) | 297(86.3) | 445(86.2) |
| | Amhara | 16(9.30) | 41(11.9) | 57(11.0) |
| | Gurage | 8(4.7) | 6(1.7) | 14(2.7) |
| **Religion** | Protestant | 89(51.7) | 179(52.0) | 268(51.9) |
| | Orthodox | 36(20.9) | 79(23.0) | 115(22.3) |
| | Catholic | 6(3.5) | 20(5.8) | 26(5.0) |
| | Muslim | 35(20.3) | 58(16.9) | 93(18.0) |
| | Others* | 6(3.5) | 8(2.3) | 14(2.7) |
| **Marital status** | Married | 156(90.7) | 319(92.7) | 475(92.1) |
| | Unmarried | 7(4.1) | 9(2.6) | 16(3.1) |
| | Others** | 9(5.2) | 16(4.7) | 25(4.8) |
| **Educational status** | Unable to read and write | 39(22.7) | 77(22.4) | 116(22.5) |
| | Completed grade 1–8 | 39(22.7) | 89(25.9) | 128(24.8) |
| | Completed grade 9–12 | 36(20.9) | 70(20.3) | 106(20.5) |
| | Diploma and above | 58(33.7) | 108(31.4) | 166(32.2) |
| **Occupation of mother** | Government employee | 40(23.3) | 94(27.3) | 134(26.0) |
| | Private employee | 38(22.1) | 50(14.5) | 88(17.1) |
| | Farmer | 43(25.0) | 99(28.8) | 142(27.5) |
| | Merchant | 47(27.3) | 84(24.4) | 131(25.4) |
| | Others*** | 4(2.3) | 17(4.9) | 21(4.1) |
| **Occupation of husband** | Government employee | 53(30.8) | 108(31.4) | 161(31.2) |
| | Private employee | 43(25.0) | 75(21.8) | 118(22.9) |
| | Merchant | 34(19.8) | 49(14.2) | 83(16.1) |
| | Farmer | 37(21.5) | 97(28.2) | 134(26.0) |
| | Others**** | 5(2.9) | 15(4.4) | 20(3.9) |
| **Residence of mother** | Rural | 116(67.4) | 214(62.2) | 330(64.0) |
| | Urban | 56(32.6) | 130(37.8) | 186(36.0) |
| **Family size** | 1–3 children | 76(44.2) | 163(47.4) | 239(46.3) |
| | 4–6 children | 83(48.3) | 146(42.4) | 229(44.4) |
| | >6 children | 13(7.6) | 35(10.2) | 48(9.3) |
| **Monthly income** | <1000 birr | 23(13.4) | 39(11.3) | 62(12.0) |
| | 1000–2000 birr | 34(19.8) | 108(31.4) | 142(27.5) |
| | 2001–3000 birr | 37(21.5) | 71(20.6) | 108(20.9) |
| | 3001–4000 birr | 41(23.8) | 57(16.6) | 98(19.0) |
| | >4000 birr | 37(21.5) | 69(20.1) | 106(20.5) |
| **Time to reach health facility** | <1 hour | 110(64.0) | 221(64.2) | 331(64.1) |
| | 1–2 hour | 31(18.0) | 79(23.0) | 110(21.3) |
| | >2 hour | 31(18.0) | 44(12.8) | 75(14.5) |

*Wakefata and non-follower of any religion

**Divorced and widowed

***House wife and daily laborer

****Student, daily laborer and student.

**Table 2. Obstetrics related characteristics of mothers attending birth at public hospitals of Western Ethiopia, 2020 (n = 516; cases: 172 and controls: 344).**

| Variables | Category | Cases N (%) | Controls N (%) | Total N (%) |
|---|---|---|---|---|
| Parity | 1st pregnancy | 34(19.8) | 91(26.5) | 125(24.2) |
| | 2 times | 8(4.7) | 12(3.5) | 20(3.9) |
| | 3 times | 13(7.6) | 22(6.4) | 35(6.8) |
| | 4 times | 52(30.2) | 90(26.2) | 142(27.5) |
| | >4 times | 65(37.8) | 129(37.5) | 194(37.6) |
| Pregnancy type | Singleton | 168(97.7) | 337(98.0) | 505(97.9) |
| | Multiple | 4(2.3) | 7(2.0) | 11(2.1) |
| Use family planning | Yes | 128(74.4) | 253(73.5) | 381(73.8) |
| | No | 44(25.6) | 91(26.5) | 135(26.2) |
| Plan of pregnancy | Yes | 121(70.3) | 278(80.8) | 399(77.3) |
| | No | 51(29.7) | 66(19.2) | 117(22.7) |
| Frequency of ANC visit | One time | 27(15.7) | 17(4.9) | 44(8.5) |
| | Two times | 22(12.8) | 52(15.1) | 74(14.3) |
| | Three times | 43(25.0) | 92(26.7) | 135(26.2) |
| | $\geq$ four times | 80(46.5) | 183(53.2) | 263(51.0) |
| Birth interval | <2 years | 54(31.4) | 56(16.3) | 110(21.3) |
| | $\geq$2 years | 118(68.6) | 288(83.7) | 406(78.7) |
| History of preterm birth | Yes | 16(9.3) | 28(8.1) | 44(8.5) |
| | No | 156(90.7) | 316(91.9) | 472(91.5) |
| History of PROM | Yes | 32(18.6) | 23(6.7) | 55(10.7) |
| | No | 140(81.4) | 321(93.3) | 461(89.3) |
| Mode of delivery | SVD | 91 (52.9) | 208 (60.5) | 299 (57.9) |
| | Forceps | 11(6.4) | 16(4.7) | 27(5.2) |
| | Cesarean section | 42(24.4) | 71(20.6) | 113(21.9) |
| | Vacuum | 22(12.8) | 46(13.4) | 68(13.2) |
| | Destructive | 6(3.5) | 3(0.9) | 9(1.7) |
| How labor started | Spontaneous | 142(82.6) | 288(83.7) | 430(83.3) |
| | Induced | 30(17.4) | 56(16.3) | 86(16.7) |
| History of abortion | Yes | 38(22.1) | 50(14.5) | 88(17.1) |
| | No | 134(77.9) | 294(85.5) | 428(82.9) |

frequency of ANC visit, birth interval, mode of delivery, plan of pregnancy, history of PROM, history of abortion, history of DM & hypertension, HIV/AIDS status, anemia during pregnancy, sexually transmitted disease, PIH, dietary supplementation during pregnancy, substance use, parity, the experience of stress, and maternal mid-upper arm circumference (MUAC) were significantly associated with the PTB at a p-value 0.25.

Variables significantly associated with PTB at P value less than 0.25 in the bivariable analysis were entered into the multivariable model. The multivariable analysis showed that lower ANC visits, short birth interval, PROM, anemia during pregnancy, PIH, and lack of dietary supplementation during pregnancy were significantly associated with the PTB. Women who had only onetime ANC attendance had 5.99 higher odds of PTB than women who attended four and above times (AOR = 5.99, 95% CI; 2.65, 13.53). Women who experienced PROM had 3.57 folds higher odds of PTB than women who do not experience PROM (AOR = 3.57, 95% CI; 1.79, 7.13). On the other hand, women who had less than a two-year birth interval had experienced 2.96 times higher odds of PTB than their counterparts (AOR = 2.96, 95% CI; 1.76, 4.98). Women who developed anemia during the current pregnancy had 4.20 folds higher odds of PTB than their counterparts (AOR = 4.20, 95% CI; 2.13, 8.28). Furthermore, the study

**Table 3. Medical illness related characteristics of mothers attending birth at public hospitals of Western Ethiopia, 2020 (n = 516; cases: 172 and controls: 344).**

| Variables | Category | Cases N (%) | Controls N (%) | Total N (%) |
|---|---|---|---|---|
| Anemia during pregnancy | Yes | 36(20.9) | 26(7.6) | 62(12.0) |
| | No | 136(79.1) | 318(92.4) | 454(88.0) |
| Malaria during pregnancy | Yes | 7(4.1) | 16(4.7) | 23(4.5) |
| | No | 165(95.9) | 328(95.3) | 493(95.5) |
| Sexually transmitted disease | Yes | 15(8.7) | 20(5.8) | 35(6.8) |
| | No | 157(91.3) | 324(94.2) | 481(93.2) |
| Pregnancy induced hypertension | Yes | 48(27.9) | 35(10.2) | 83(16.1) |
| | No | 124(72.1) | 309(89.8) | 433(83.9) |
| History of DM | Yes | 7 (4.1) | 28 (8.1) | 35 (6.8) |
| | No | 165 (95.9) | 316 (91.9) | 481(93.2) |
| History of hypertension | Yes | 35 (20.3) | 50 (14.5) | 85 (16.5) |
| | No | 137 (79.7) | 294 (85.5) | 431(83.5) |
| History of cardiac disease | Yes | 5 (2.9) | 5 (1.5) | 10 (1.9) |
| | No | 167 (97.1) | 339 (98.5) | 506 (98.1) |
| HIV/AIDS status | Positive | 5 (2.9) | 1(0.3) | 6 (1.2) |
| | Negative | 156 (90.7) | 336 (97.7) | 492 (95.3) |
| | Unknown | 11(6.4) | 7 (2.0) | 18 (3.5) |

found that women who didn't get dietary supplementation during pregnancy had 2.43 higher odds of PTB than their counterparts (AOR = 2.43, 95% CI; 1.51, 3.91). The study also revealed that mothers who developed PIH were 3.13 times higher odds of developing PTB than their counterparts (AOR = 3.13, 95% CI; 1.78, 5.50) (Table 4).

## Discussions

PTB remains to be a global agenda as its complication accounts for 35% of neonatal death worldwide. There is a paucity of data on risk factors of PTB in Ethiopia, particularly in the western part of the country.

The current study found a short birth interval as a risk factor for PTB. This finding is consistent with the study conducted in the Amhara region [29], Jimma Medical center [36], Axum and Adwa town [17]. It is also supported by a meta-analysis of eight studies that found pregnancy intervals of < 6 months were associated with PTB compared with pregnancy intervals of 18–23 months [43]. Another study conducted in the USA on adolescent women also reported a similar finding with our current study [44]. However, the finding of this study contrast with a study done in Debretabor town, Sidama zone, and Kenya [12, 18, 35] where the birth interval was not associated with preterm delivery. This finding implies that there is a need to focus on increasing access to and use of contraception for potentially delaying the second pregnancy. Increasing mothers' use of effective contraception can address both unintended births and PTB.

The current study also pointed out the PROM as a risk factor of PTB. This finding is supported by a study done in Amhara Region Referral Hospitals [31], Sidama Zone [35], Jimma medical center [36], Debretabor town [18], and Axum and Adwa town [17]. Some of the studies conducted in African countries including Kenya [12] and Ghana [33] also support the current study finding. This finding is contrary finding with similar studies on preterm delivery, which observed that PROM was not associated with pre-term delivery [16, 32]. PROM can lead to uterine contraction as amniotic fluid contains prostaglandin which in turn may result in PTB. Besides, PROM elevates fetal plasma interleukin-6 which may trigger preterm labor

**Table 4. Bivariable and multivariable logistic regression analysis of PTB among women who gave birth at public hospitals of western Ethiopia, 2020 (n = 516; cases: 172 and controls: 344).**

| Variables | | Preterm birth | | COR (95% CI) | AOR (95% CI) | P-Value |
|---|---|---|---|---|---|---|
| | | Cases (%) | Controls (%) | | | |
| Age at marriage | <18 years | 18(10.5) | 50(14.5) | 0.57(0.254, 1.287) | 0.35(0.12, 1.03) | 0.058 |
| | 18–23 years | 137(79.7) | 267(77.6) | 0.82(0.429, 1.547) | 1.00(0.46, 2.17) | 0.99 |
| | >23 years | 17(9.9) | 27(7.8) | 1 | 1 | |
| Ethnicity | Oromo | 148(86.0) | 297(86.3) | 1 | 1 | |
| | Amhara | 16(9.30) | 41(11.9) | 0.78(0.425, 1.442) | 1.06(0.43, 2.61) | 0.89 |
| | Gurage | 8(4.7) | 6(1.7) | 2.68(0.912, 7.853) | 1.50(0.33, 6.83) | 0.60 |
| Age of mother | 15–24 years | 50(29.1) | 87(25.3) | 0.95(0.56, 1.62) | 1.39(0.70, 2.79) | 0.34 |
| | 25–34 years | 83(48.3) | 192(55.8) | 0.72(0.45, 0.93) | 0.79(0.44, 1.43) | 0.45 |
| | ≥35 years | 39(22.7) | 65(18.9) | 1 | 1 | |
| Husband's occupation | Gov't employee | 53(30.8) | 108(31.4) | 1 | 1 | |
| | Private employee | 43(25.0) | 75(21.8) | 1.17(0.710, 1.924) | 1.21(0.67, 2.20) | 0.52 |
| | Merchant | 34(19.8) | 49(14.2) | 1.41(0.818, 2.444) | 1.29(0.67, 2.51) | 0.44 |
| | Farmer | 37(21.5) | 97(28.2) | 0.78(0.471, 1.283) | 0.48(0.25, 0.95) | 0.056 |
| | Others | 5(2.9) | 15(4.4) | 0.680(.234, 1.969) | 0.97(0.26, 3.62) | 0.96 |
| Residence of mother | Rural | 116(67.4) | 214(62.2) | 1 | 1 | |
| | Urban | 56(32.6) | 130(37.8) | 0.80(0.540, 1.169) | 0.83(0.40, 1.71) | 0.62 |
| Mothers occupation | Gov't employee | 40(23.3) | 94(27.3) | 1 | 1 | |
| | Private employee | 38(22.1) | 50(14.5) | 1.79(1.019, 3.130) | 1.83(0.84, 3.98) | 0.12 |
| | Farmer | 43(25.0) | 99(28.8) | 1.02(0.610, 1.708) | 2.03(0.68, 6.11) | 0.20 |
| | Merchant | 47(27.3) | 84(24.4) | 1.32(0.786, 2.199) | 1.27(0.60, 2.68) | 0.53 |
| | Others | 4(2.3) | 17(4.9) | 0.55(0.175, 1.747) | 0.55(0.13, 2.36) | 0.42 |
| Monthly income | <1000 birr | 23(13.4) | 39(11.3) | 1.10(0.573, 2.111) | 1.72(0.71, 4.16) | 0.22 |
| | 1000–2000 birr | 34(19.8) | 108(31.4) | 0.59(0.337, 1.023) | 0.65(0.32, 1.32) | 0.23 |
| | 2001–3000 birr | 37(21.5) | 71(20.6) | 0.97(0.553, 1.707) | 1.21(0.60, 2.42) | 0.58 |
| | 3001–4000 birr | 41(23.8) | 57(16.6) | 1.34(0.761, 2.363) | 1.53(0.77, 3.01) | 0.21 |
| | >4000 birr | 37(21.5) | 69(20.1) | 1 | 1 | |
| Time to reach health facility | <1 hour | 110(64.0) | 221(64.2) | 1 | 1 | |
| | 1–2 hour | 31(18.0) | 79(23.0) | 0.79(.491, 1.267) | 1.14(0.61, 2.10) | 0.67 |
| | >2 hour | 31(18.0) | 44(12.8) | 1.42(0.847, 2.365) | 1.53(0.74, 3.16) | 0.24 |
| Frequency of ANC visit | One time | 27(15.7) | 17(4.9) | 3.63(1.875, 7.038) | 5.99(2.65, 13.53) | 0.000* |
| | Two times | 22(12.8) | 52(15.1) | 0.97(.551, 1.700) | 0.76(0.37, 1.57) | 0.46 |
| | Three times | 43(25.0) | 92(26.7) | 1.07(0.684, 1.672) | 0.95(0.54, 1.66) | 0.86 |
| | ≥ four times | 80(46.5) | 183(53.2) | 1 | 1 | |
| Birth interval | <2 years | 54(31.4) | 56(16.3) | 2.35(1.530, 3.621) | 2.96(1.76, 4.98) | 0.000* |
| | ≥2 years | 118(68.6) | 288(83.7) | 1 | 1 | |
| Mode of delivery | SVD | 91 (52.9) | 208 (60.5) | 1 | 1 | |
| | Forceps | 11(6.4) | 16(4.7) | 1.57(0.702, 3.519) | 0.99(0.34, 2.88) | 0.98 |
| | CS | 42(24.4) | 71(20.6) | 1.35(0.859, 2.129) | 0.85(0.46, 1.57) | 0.61 |
| | Vacuum delivery | 22(12.8) | 46(13.4) | 1.09(0.622, 1.923) | 0.70(0.33, 1.45) | 0.34 |
| | Destructive | 6(3.5) | 3(0.9) | 4.57(1.119, 18.68) | 4.14(0.79, 21.67) | 0.09 |
| Plan of pregnancy | Yes | 121(70.3) | 278(80.8) | 1 | 1 | |
| | No | 51(29.7) | 66(19.2) | 1.78(1.163, 2.711) | 1.42(0.77, 2.61) | 0.25 |
| History of PROM | Yes | 32(18.6) | 23(6.7) | 3.19(1.802, 5.649) | 3.57(1.79, 7.13) | 0.000* |
| | No | 140(81.4) | 321(93.3) | 1 | 1 | |

(*Continued*)

**Table 4.** (Continued)

| Variables | | Preterm birth | | COR (95% CI) | AOR (95% CI) | P-Value |
|---|---|---|---|---|---|---|
| | | Cases (%) | Controls (%) | | | |
| History of Abortion | Yes | 38(22.1) | 50(14.5) | 1.67(1.044, 2.664) | 1.08(0.29, 3.44) | 0.99 |
| | No | 134(77.9) | 294(85.5) | 1 | 1 | |
| History of DM | Yes | 7 (4.1) | 28 (8.1) | 0.48(0.205, 1.119) | 0.33(0.11, 0.92) | 0.064 |
| | No | 165(95.9) | 316(91.9) | 1 | 1 | |
| History of hypertension | Yes | 35 (20.3) | 50 (14.5) | 1.50(.932, 2.421) | 1.21(0.62, 2.37) | 0.56 |
| | No | 137 (79.7) | 294 (85.5) | 1 | 1 | |
| HIV/AIDS Status | Positive | 5 (2.9) | 1(0.3) | 1 | 1 | |
| | Negative | 156 (90.7) | 336 (97.7) | 0.09(0.011, 0.801) | 1.59(0.12, 20.34) | 0.72 |
| | Unknown | 11(6.4) | 7 (2.0) | 0.31(0.030, 3.285) | 1.97(0.11, 32.94) | 0.63 |
| Anemia during this pregnancy | Yes | 36(20.9) | 26(7.6) | 3.24(1.881, 5.572) | 4.20(2.13, 8.28) | 0.000* |
| | No | 136(79.1) | 318(92.4) | 1 | 1 | |
| Sexually transmitted disease | Yes | 15(8.7) | 20(5.8) | 1.55(0.772, 3.105) | 2.34(0.97, 5.63) | 0.058 |
| | No | 157(91.3) | 324(94.2) | 1 | 1 | |
| Pregnancy induced hypertension | Yes | 48(27.9) | 35(10.2) | 3.42(2.109, 5.539) | 3.13(1.78, 5.50) | 0.000* |
| | No | 124(72.1) | 309(89.8) | 1 | 1 | |
| Dietary supplementation | Yes | 90(52.3) | 226(65.7) | 1 | 1 | |
| | No | 82(47.7) | 118(34.3) | 1.75(1.202, 2.534) | 2.43(1.51, 3.91) | 0.000* |
| History of substance use | Yes | 36(20.9) | 42(12.2) | 1.90(1.167, 3.104) | 1.74(0.97, 3.12) | 0.060 |
| | No | 136(79.1) | 302(87.8) | 1 | 1 | |
| Experienced stress | Yes | 42(24.4) | 66(19.2) | 1.36(0.877, 2.112) | 0.98(0.52, 1.84) | 0.95 |
| | No | 130(75.6) | 278(80.8) | 1 | 1 | |
| Parity | 1st pregnancy | 34(19.8) | 91(26.5) | 1 | 1 | |
| | 2 times | 8(4.7) | 12(3.5) | 1.78(0.671, 4.742) | 2.74(0.80, 9.31) | 0.10 |
| | 3 times | 13(7.6) | 22(6.4) | 1.58(0.717, 3.487) | 2.03(0.76, 5.46) | 0.15 |
| | 4 times | 52(30.2) | 90(26.2) | 1.55(0.918, 2.604) | 2.37(1.15, 4.88) | 0.19 |
| | >4 times | 65(37.8) | 129(37.5) | 1.35(0.823, 2.210) | 2.15(1.09, 4.26) | 0.27 |
| Maternal MUAC | ≥23cm | 106(61.6) | 245(71.2) | 1 | 1 | |
| | <23 cm | 66(38.4) | 99(28.8) | 1.54(1.048, 2.267) | 1.00(0.49, 2.07) | 0.98 |

*shows significant at P-value <0.05.

COR: AOR: Adjusted Odd Ratio, CI: Confidence Interval.

and leads to preterm delivery. PROM is also associated with lower latency from membrane rupture until delivery and causes around 25–30% of all preterm deliveries [45].

Furthermore, this study revealed that PIH is another risk factor of PTB. PIH can cause vascular damage to the placenta causing abruption placenta which results in PTB. Also, uteroplacental ischemia is a plausible explanation for the PTB associated with PIH. When the blood pressure becomes uncontrollable, the quickest means of emptying the uterus become the choice accounting for the preterm delivery. In addition, elevated blood pressure in pregnancy compromises perfusion to the fetus and has a medical risk of cardiovascular complications for the mother. This finding is in line with the existing evidence from Amhara Region Referral Hospitals [31], Gondar town [32], Addis Ababa [16], Ghana [33], Central zone of Tigray [34], Kenya [12], Sidama zone [35] and Jimma medical center [36]. Literature from China also indicated that PIH is a significant risk factor for PTB and that the risk of PTB is higher among women with PIH [46]. However, this finding is not supported by some of the previously conducted studies [18, 19, 37].

This study finding also showed that the frequency of ANC visits was associated with PTB. This finding is consistent with the study done in Ghana [33], the Central zone of Tigray [34], Amhara region [29], Dodola town [37], Jimma medical center [36], and Debretabor town [18]. Some of the studies conducted in African countries including Ethiopia [12, 17] didn't show an association between the frequency of ANC visits and PTB. The less frequent a mother visits for ANC; the late obstetric problems are identified which as a result end with PTB. It is also known that ANC visits during pregnancy help to monitor the wellbeing of the fetus. Besides, higher ANC visit maximizes the opportunity for early identification and treatment of obstetric complications. Therefore, the lack of adequate ANC visits during pregnancy decreases the chance of identifying risks of PTB and providing appropriate interventions for its prevention.

Lack of dietary supplementation during pregnancy was another independent predictor of PTB in the present study. This finding is similar to the results of the study conducted in Tigray [38] and Debretabor town [18]. However, the studies conducted in Dodola town [37], Jimma medical center [36], Ghana [33], and Central zone of Tigray [34] show no association of dietary supplementation with PTB. When the mother's nutritional status is poor; they will be prone to chronic infection which may lead to the activation of the maternal-fetal immune system causing preterm labor.

Moreover, the study identified anemia as a risk factor of PTB, and this finding is consistent with the study conducted in Shire Sihul [19], Debretabor town [18], Amhara Region Referral Hospitals [31], and Axum and Adwa town [17]. This finding is also supported by the study conducted in Malawi and Indonesia [47, 48]. However, this study findings contrast with a study done in Sidama zone [35], Tigray [34], Addis Ababa [16], Ghana [33], and Kenya [12] where the presence of anemia during pregnancy was not associated with preterm delivery. Anemia may lead to decreased blood flow to the placenta and results in preterm labor which in turn results in PTB. Also, anemia may cause hypoxia which can induce fetal stress, which stimulates the production of the corticotrophin-releasing hormone (CRH) leading to preterm labor. Iron deficiency may also increase the risk of maternal infections which can again stimulate the production of CRH predisposing to PTB. Even though available evidence suggests multiple pregnancies, history of abortion, and history of PTB as an independent predictor of PTB, all of them showed no statistically significant association with PTB in the current study [17, 19, 29, 31, 34–36].

## Limitation of the study

There may be recall bias from mothers due to the nature of some question which dealt with past information. To reduce this recall bias, information gathered from the mothers through the interview was cross-checked from their antenatal records. Due to the sensitive nature of some questions and face-to-face techniques of data collection, there is a possibility of falsified reporting (social desirability bias) among mothers. We made an effort to minimize this by assuring mothers for the confidentiality of their information. Another limitation of the study is being an institution-based quantitative study. It would be better if a community-based and qualitative approach study was triangulated with the quantitative part to investigate further factors on PTB. On the other hand, the strength of this study is we used a strong design with a large sample size with a 1:2 ratio of cases to controls.

## Conclusions

ANC service providers should focus on mothers with PIH, PROM, and anemia during pregnancy, and refer to the senior experts for early management to reduce the risk of preterm delivery. ANC services such as counseling the mother on the benefit of dietary

supplementation during pregnancy, ANC follow up, and lengthening birth interval should be integrated into the existing health extension packages. Emphasizing these determinants with appropriate care during pregnancy is essential to reduce the occurrence of PTB. Increasing the awareness of contraceptive utilization and counseling to enhance birth spacing, ANC visits, folic acid, and dietary supplementation during pregnancy should be given strict attention by healthcare providers. New and inclusive strategies such as the establishment of comprehensive mobile clinic services should also be designed to reduce the burden of PTB among women living in the rural community. Lastly, we recommend future researchers to conduct longitudinal and community-based studies supplemented with qualitative methods.

## Supporting information

**S1 Questionnaire.**
(DOCX)

## Acknowledgments

We would like to acknowledge the Wollega zone health bureau and each hospital administrative office for their cooperation. We are also grateful to the study participants who voluntarily agreed to be interviewed and participated in the study.

## Author Contributions

**Conceptualization:** Muktar Abadiga, Bizuneh Wakuma, Adugna Oluma, Ginenus Fekadu, Nesru Hiko, Getu Mosisa.

**Data curation:** Muktar Abadiga, Adugna Oluma, Ginenus Fekadu, Nesru Hiko.

**Formal analysis:** Muktar Abadiga, Bizuneh Wakuma, Adugna Oluma, Getu Mosisa.

**Funding acquisition:** Muktar Abadiga, Bizuneh Wakuma, Ginenus Fekadu, Nesru Hiko, Getu Mosisa.

**Investigation:** Muktar Abadiga, Adugna Oluma, Nesru Hiko.

**Methodology:** Muktar Abadiga, Bizuneh Wakuma, Adugna Oluma, Ginenus Fekadu, Nesru Hiko, Getu Mosisa.

**Project administration:** Muktar Abadiga, Adugna Oluma, Getu Mosisa.

**Resources:** Muktar Abadiga, Bizuneh Wakuma, Adugna Oluma, Ginenus Fekadu, Nesru Hiko, Getu Mosisa.

**Software:** Muktar Abadiga.

**Supervision:** Muktar Abadiga, Bizuneh Wakuma, Adugna Oluma, Ginenus Fekadu, Nesru Hiko, Getu Mosisa.

**Validation:** Muktar Abadiga, Adugna Oluma, Ginenus Fekadu, Nesru Hiko.

**Visualization:** Muktar Abadiga, Bizuneh Wakuma, Adugna Oluma, Ginenus Fekadu, Getu Mosisa.

**Writing – original draft:** Muktar Abadiga, Bizuneh Wakuma, Adugna Oluma, Ginenus Fekadu, Nesru Hiko, Getu Mosisa.

**Writing – review & editing:** Muktar Abadiga, Bizuneh Wakuma, Adugna Oluma, Ginenus Fekadu, Nesru Hiko, Getu Mosisa.

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
