## [Decision Letter · Decision Letter 0]

14 Sep 2020

PONE-D-20-17494

Determinants of preterm birth among women delivered in public hospitals of Western Ethiopia, 2020: Unmatched case-control study

PLOS ONE

Dear Dr. Abadiga,

Thank you for submitting your manuscript to PLOS ONE. After careful consideration, we feel that it has merit but does not fully meet PLOS ONE’s publication criteria as it currently stands. Therefore, we invite you to submit a revised version of the manuscript that addresses the points raised during the review process.

We look forward to receiving your revised manuscript.

Kind regards,

Florian Fischer

Academic Editor

PLOS ONE

Journal Requirements:

2. In the manuscript text, please provide additional details regarding participant consent. Please ensure that you have specified (1) whether consent was informed and (2) what type you obtained (for instance, written or verbal, and if verbal, how it was documented and witnessed). If your study included minors, state whether you obtained consent from parents or guardians.

3.Please include additional information regarding the survey or questionnaire used in the study and ensure that you have provided sufficient details that others could replicate the analyses. For instance, if you developed a questionnaire as part of this study and it is not under a copyright more restrictive than CC-BY, please include a copy, in both the original language and English, as Supporting Information.

In addition, in the Methods, please describe how the questionnaire was pre-tested and/or validated. If this did not occur, please provide the rationale for not pre-testing or validating the questionnaire.

Reviewers' comments:

Reviewer's Responses to Questions

**Comments to the Author**

1. Is the manuscript technically sound, and do the data support the conclusions?

Reviewer #1: Partly

Reviewer #2: Yes

2. Has the statistical analysis been performed appropriately and rigorously? 

Reviewer #1: Yes

Reviewer #2: No

3. Have the authors made all data underlying the findings in their manuscript fully available?

Reviewer #1: Yes

Reviewer #2: Yes

4. Is the manuscript presented in an intelligible fashion and written in standard English?

Reviewer #1: No

Reviewer #2: Yes

5. Review Comments to the Author

Reviewer #1: Abstract

It is good but the result and the conclusions needs a revision. The result does not show the association in clearly and the direction of association is not clear. In the conclusion, it lacks bold and clear suggestion for the policy and programmatic startegic implication of the study.

Background

The introduction has well written about the contexts. However, it lacks some depth about the reason why the need of conducting this study and in this section, the knowledge gap was not clearly explained clearly, how would the study help the country to increase attentions in PTB prevention and control strategies? At the same time, the literature the authors mentioned some research conducted on similar topic in the same country, method with only difference in districts. Hence, the researcher needs to justify and the passion of doing this as an additional interest.

Methods

• The researcher needs to mention and show the sampling done and the potential selection bias that might be introduced in the methodology.

• In the analysis and model fitting-, how the researcher manage the small observation in the outcome variable particularly in the multiple regression is not well explained this might affect the final picture and findings of the findings. Therefore the researcher needs to clearly show this problem in the method section.

Discussion

1. The discussion at times looks a replica of the result and just bring the other studies that looks alike. However, please focused and build on mainly the justification for those differences and learning aspects from the findings

2. At the same time, the research needs to clearly state where this research limitation and strength. How those issues resolved

Conclusion and recommendation

Can the researcher show clearly, the significance to the public health and policy or implication in broader perspective. What are importance and the contribution from the existing system? can give concert suggestions to make a change based on the evidence of their study.

Reviewer #2: Review Report for manuscript number “PONE-D-20-17494”

1) General comments: This study is relevant as it was conducted on preterm birth which is one of the leading causes of neonatal deaths in Ethiopia. However, it needs language revisions throughout the manuscript. Moreover, major changes should be made by taking the provided comments into account.

2) Specific Comments:

Abstract:

A) Background: Line 27 states that “There were few studies on determinants of preterm birth in Ethiopia…”. I don’t think that. There are a number of studies in Ethiopia in this issue. You would rather mention their gaps.

B) Method: Line 37; the term “Data” is plural word. So, replace “was” with “were”.

C) Conclusion: please propose certain recommendations based on your key findings.

Introduction

A) I kindly recommend you to summarize the statement of the problem with 4 paragraphs: 1) the nature of the problem, 2) the known aspects of the problem (causes, magnitude risk factors, intervention (i.e., what is known?)), 3) the unknown aspects (any knowledge, intervention, methodology gaps (i.e., what is unknown?)), and 4) the expectation (s) from this study (i.e., so what?)

B) Please work out to improve the coherence.

C) I kindly request you for clarification: Line 55-57; it has to be clear in which regions or countries this classification works? For Example, in Ethiopia context, any termination of pregnancy prior to 28 completed weeks of gestational age is termed as "abortion".

D) Please correct flaring errors. For example, line 58 states that “about 15 million infant born prematurely.” Herein, please change the infant to its plural form (i.e., infants). Ditto for line 86; change “literatures” to literature.

E) Line 64-66; could you please provide information on the aggregate prevalence of preterm birth in Ethiopia.

Methods:

A) Line 123-125; generally further elaborations would be included regarding the case definition, inclusion and exclusion criteria. The cases and controls would be defined clearly, ‘’ not reliable ultrasonography’ has to be measurable. What measures have been made when gestational age disparity happened across records from LNMP and “early ultrasound”. Have you included post term births to controls? Have you included those preterm births which have been terminated deliberately? In general, ACOG definition would be used.

B) Line 128-132; It is not clear why the authors took reference from Ghana for sample size calculation as a number of studies are here in Ethiopia (For instance; look at https://doi.org/10.1155/2020/1854073).

Result:

A) Line 198-199; “Thirty-nine (22.67%) of cases and 77 (22.38%) of controls had no formal education” please add the article “the” in before “cases” &” controls”

B) Line 201-202; “Concerning residence, about two-third of the cases 116 (67.44%) and 214 (62.21%) of the controls were rural dwellers” would be corrected as one of the following:

1) Concerning residence, about 116 (67.44%) of the cases and 214 (62.21%) of the controls were rural dwellers or 2) Concerning residence, about two-third (67.44%) of the cases and three-fifth (62.21%) of the controls were rural dwellers.

C) Line 208; “… majority of the study participants, 128 (74.42%) of the cases 253 (73.55%) of the controls …” needs English language editing.

D) The regression table (i.e., “table 4”) poses major issues:

One category of occupation is “others”, what “others” stands for? List them by using foot note.

One category of “Marital status” is “widowed”. However, the number of cases in this category is only one and the corresponding 95% CI is too wide (i.e., 0.029, 1.809). In other words, about 1780 differences observed between the upper and lower limit which violates one of the assumptions of the binary logistic regression model. So, I kindly recommend you to merge the some categories of the variables with few numbers of participants in the cases and / or controls. Thereafter, you need to undertake re-analysis provided that all variables which are entered to the model should fulfill the assumptions of the Binary logistic regression model.

About 50 cases and 87 controls were in the age group of 15-24 which implies that minors have been included in the study. If so, how have you addressed the ethical issues? In this occasion, let me raise one another important concern; why have you failed in incorporating a sub-section of “Ethical approval”?

The sum of each category of variables under the cases has to be 100%. Ditto for controls. However, that was not true in your study. This implies that you have considered the “row” percentages in the “crosstab” that is recommended for cross sectional study design. For case – control study design, however, “column” percentages should be reported over “row” percentages. Therefore, you need to address this concern while you perform re-analysis as per the above recommendation.

Discussion:

Your way of discussion is interesting. However, still it needs revision for language:

A) Tense or spelling errors: For instance; line 274; change “remained” to be “remains”, line 309; edit “Gonder” to be “Gondar”

B) Certain phrases have been employed frequently. For example “the odds of developing preterm birth...”, “…due to the fact that…”

C) Be consistent in using the abbreviations Vs the extended forms. For example; line 302; you have utilized “Pregnancy induced hypertension”, whereas at line 303; you have used its abbreviation for “PIH”, again at line 305, 307, 311…; you have employed its extended form “Pregnancy induced hypertension”???Moreover, the abbreviation “PIH “has not been listed under the “Abbreviations” section. On the contrary, the abbreviation “PIHTN” has been found under the list although it has not been used at the main document at all. I kindly recommend you to put the extended form of each abbreviation followed by its abbreviated form in bracket at its first appearance. Then you can use the abbreviated form alone throughout the document and don’t forget mentioning it at the lists of “abbreviations “section.

D) The limitations of your study as well as the efforts made to overcome those limitations would be stated.

References:

A) Reference 6(line 374-376) & 28 (line 430-432) are similar.

B) You would use certain recently published articles. For example “Determinants of Preterm Birth among Women Who Gave Birth in Amhara Region Referral Hospitals, Northern Ethiopia, 2018: Institutional Based Case Control Study”, which is available at https://doi.org/10.1155/2020/1854073 , has not been cited in your manuscript.

6. PLOS authors have the option to publish the peer review history of their article (what does this mean?). If published, this will include your full peer review and any attached files.

Reviewer #1: No

Reviewer #2: **Yes: **Muhabaw Shumye Mihret

---

## [Author Response · Author response to Decision Letter 0]

28 Sep 2020

Point by point response

 Journal Requirements:

Response: yes, we made our manuscript meet PLOS ONE's style requirements.

2. In the manuscript text, please provide additional details regarding participant consent. Please ensure that you have specified (1) whether consent was informed and (2) what type you obtained (for instance, written or verbal, and if verbal, how it was documented and witnessed). If your study included minors, state whether you obtained consent from parents or guardians.

Response: Sorry for not including ethics and participant consent in the previous version of our manuscript. We included all these informations in this revised version of our manuscript as follows: 

“Ethics approval and consent to participate

The study was approved by the institutional review boards of Wollega University ethical review board with approval ID: HIS/213/20. Permission letter was also obtained from each hospital administrative office. All participants of the study were provided written consent, clearly stating the objectives of the study and their right to refuse. Then, written informed consent was obtained from the study participants. For minors, informed consent was received from their parents or legal guardians. To ensure confidentiality, names or identifying information was not indicated on the questionnaires. Mothers were interviewed in private rooms to ensure their privacy. The filled questionnaires were carefully handled ensuring confidentiality and was kept under secured custody of the corresponding author”.

Response: The questionnaire used in this study was adapted from the Ethiopian Demographic and Health Survey and other similar studies. The necessary modifications were done to be applicable to the current study and population. We included this information in the methods part of this revised manuscript. The reference from which this questionnaire was adapted were indicated in brackets. The English version of this questionnaire is also provided as supplementary file in this revised submission. 

4. In addition, in the Methods, please describe how the questionnaire was pre-tested and/or validated. If this did not occur, please provide the rationale for not pre-testing or validating the questionnaire.

Response: Five percent of the questionnaire (32 study participants) was pre-tested 5 days prior to data collection and modification was made based on pre-test result. This information is included in this revised manuscript under “data quality control”. 

 Reviewers' comments:

Reviewer #1: 

Abstract

It is good but the result and the conclusions needs a revision. The result does not show the association in clearly and the direction of association is not clear.

Response: We re-wrote the result in a way it can show direction of association in this revised manuscript. We re-wrote as follows: “Multivariable analysis show that mothers who developed pregnancy-induced hypertension (AOR=3.16, 95% CI; 1.800, 5.542), only one time ANC visits (AOR=5.88, 95% CI; 2.570, 13.454), experienced premature rupture of membrane (AOR=3.75, 95% CI; 1.872, 7.511), less than two year birth interval (AOR=3.10, 95% CI; 1.832, 5.248), developed anemia during the current pregnancy (AOR=4.60, 95% CI; 2.338, 9.064) and didn’t get dietary supplementation during pregnancy (AOR=2.41, 95% CI; 1.489, 3.911) had statistically significant association with experiencing preterm birth”.

 In the conclusion, it lacks bold and clear suggestion for the policy and programmatic strategic implication of the study.

Response: Based on your request, we gave bold suggestion in this revised manuscript. The suggestion given is as follows: “Antenatal care service providers should focus on mothers with pregnancy-induced hypertension, premature rupture of membrane and anemia during pregnancy, and refer to the senior experts for early management in order to reduce risk of preterm delivery. Antenatal care services such as counseling the mother on the benefit of dietary supplementation during pregnancy, antenatal care follow up and lengthening birth interval should be integrated into the existing health extension packages. New and inclusive strategies such as establishment of comprehensive mobile clinic services should also be designed to reduce the burden of preterm birth among women living in the rural community. Lastly, we recommend future researchers to conduct longitudinal and community-based studies supplemented with qualitative methods”.

Background

The introduction has well written about the contexts. However, it lacks some depth about the reason why the need of conducting this study and in this section, the knowledge gap was not clearly explained clearly, how would the study help the country to increase attentions in PTB prevention and control strategies? At the same time, the literature the authors mentioned some research conducted on similar topic in the same country, method with only difference in districts. Hence, the researcher needs to justify and the passion of doing this as an additional interest.

Response: We included the following information to justify our study. “Due to the enormous economic and emotional burden of preterm birth, identifying the risk factors for preterm birth has the potential to help in preventing the impacts. It is very important to understand the risk factors of preterm birth especially in developing countries in which the rate of preterm birth is high. However, no study was conducted on determinants of preterm birth in western part of a country. Due to variation in socioeconomic status and health care service coverage, the risk factors of preterm birth may vary from region to region and time trends even within the same country. On the other hand, most of the studies conducted in other parts of a country were based on card review and therefore, the risk factors of preterm birth were not fully addressed. Hence, only limited risk factors of preterm birth were assessed in the previous study conducted in Ethiopia. On the contrary, this study was based on primary data and included many risk factors which would be associated with preterm birth. In addition, most of the studies conducted in Ethiopia were cross-sectional and our study used case-control study design which is stronger than cross-sectional. Therefore, this study was aimed to identify determinants of preterm birth in western part of Ethiopia using strong design. The finding of this study is important for policymakers and health care workers by providing important information related to risk factors of preterm birth in designing an effective strategy to prevent and control preterm birth. All these informations are included in this revised manuscript. 

Methods

• The researcher needs to mention and show the sampling done and the potential selection bias that might be introduced in the methodology.

Response: First, six hospitals found in Wollega zones were randomly selected, namely; Nedjo general hospital, Gimbi general hospital, Nekemte specialized hospital, Arjo hospital, Wollega university specialized hospital and Shambu general hospital. Then, the number of cases and controls were proportionally allocated to each hospital based on the number of mothers who gave birth at each selected hospital within four months prior to the data collection time. Finally, the eligible case was selected consecutively and the consecutive two controls were selected until the required sample size was achieved. Accordingly, we included 34 cases and 68 controls from Nekemte specialized Hospital, 32 cases and 64 controls from Wollega university specialized hospital, 20 cases and 40 controls from Arjo Hospital, 43 cases and 86 controls from Gimbi general hospital, 29 cases and 58 controls from Shambu Hospital and 30 cases and 60 controls from Nedjo hospital. The interview was held in a separate room after a woman is stabilized and ready to be discharged. These all informations are included in this revised version of our manuscript.

• In the analysis and model fitting-, how the researcher manages the small observation in the outcome variable particularly in the multiple regression is not well explained this might affect the final picture and findings of the findings. Therefore, the researcher needs to clearly show this problem in the method section.

Response: Thank you very much for this important comment. Some of the categories of our variables have small observation in the outcome variable. For example: mothers age (category ‘>44’ years) and marital status (category ‘widowed’) have small observations. We didn’t account this problem during the previous analysis of our data. Therefore, based on your comments, we merged some categories of the variables with few numbers of participants in the cases and / or controls and undertake re-analysis to fulfill the assumptions of the Binary logistic regression. So, there are some differences in finding of cross-tabulation, AOR, CI and P value between the former and this revised manuscript. These measures taken were explained in the method section of this revised manuscript.

Discussion

1. The discussion at times looks a replica of the result and just bring the other studies that looks alike. However, please focused and build on mainly the justification for those differences and learning aspects from the findings

Response: We included many results which contradicts our finding and justification for the differences in this revised manuscript. Thank you for this important comment.

2. At the same time, the research needs to clearly state where this research limitation and strength. How those issues resolved

Response: We forgot including limitation of the study in the previous submission. We included limitation of this study in this revised submission as follows: “There may be recall bias from mothers due to the nature of some question which dealt with past informations. To reduce this recall bias, information gathered from the mothers through the interview were crosschecked from their antenatal records. Due to the sensitive nature of some questions and face-to-face techniques of data collection, there is a possibility of falsified reporting (social desirability bias) among mothers. We made an effort to minimize this by assuring mothers for the confidentiality of their informations. Another limitation of the study is being institution-based quantitative study. It would be better if community-based and qualitative approach study was triangulated with quantitative part to investigate further factors on preterm birth. On the other hand, the strength of this study is we used strong design with large sample size with 1:2 ratio of cases to controls”.

Conclusion and recommendation

Can the researcher show clearly, the significance to the public health and policy or implication in broader perspective? What are importance and the contribution from the existing system? can give concert suggestions to make a change based on the evidence of their study.

Response: We agree that our conclusion lacks bold suggestion for health care provider and policy makers. However, we gave bold and clear suggestion in this revised submission. The suggestion we gave in this revised submission is as follows: “Antenatal care service providers should focus on mothers with pregnancy-induced hypertension, premature rupture of membrane and anemia during pregnancy, and refer to the senior experts for early management in order to reduce risk of preterm delivery. Antenatal care services such as counseling the mother on the benefit of dietary supplementation during pregnancy, antenatal care follow up and lengthening birth interval should be integrated into the existing health extension packages. Giving emphasis to these determinants with appropriate care during pregnancy is essential to reduce the occurrence of preterm birth. Increasing the awareness of contraceptive utilization and counseling to enhance birth spacing, antenatal care visits, folic acid and dietary supplementation during pregnancy should be given strict attention by healthcare providers. New and inclusive strategies such as establishment of comprehensive mobile clinic services should also be designed to reduce the burden of PTB among women living in the rural community. Lastly, we recommend future researchers to conduct longitudinal and community-based studies supplemented with qualitative methods”.

Reviewer #2: 

1) General comments: This study is relevant as it was conducted on preterm birth which is one of the leading causes of neonatal deaths in Ethiopia. However, it needs language revisions throughout the manuscript. Moreover, major changes should be made by taking the provided comments into account.

Response: Thank you very much for your important comments. The previous version of our manuscript has some grammatical and editorial problems. All authors intensively reviewed the document and corrected grammatical and editorial problems. We also used free online grammar correction to solve this problem in this revised version of our manuscript. We also contacted an expert to edit the English in this manuscript.

2) Specific Comments:

Abstract:

A) Background: Line 27 states that “There were few studies on determinants of preterm birth in Ethiopia…”. I don’t think that. There are a number of studies in Ethiopia in this issue. You would rather mention their gaps.

Response: We agree with a reviewer that mentioning the gaps of previous study is better to justify one study. Therefore, we justified in this revised submission as follows: “It is very important to understand the risk factors of preterm birth especially in developing countries in which the rate of preterm birth is high. However, though there were numerous studies in Northern, southern and eastern part of Ethiopia, no study was conducted on determinants of preterm birth in western part of a country. Due to variation in socioeconomic status and health care service coverage, the risk factors of preterm birth may vary from region to region and time trends even within the same country. On the other hand, most of the studies conducted in other parts of a country were based on card review, and therefore, the risk factors of preterm birth were not fully addressed. Hence, only limited risk factors of preterm birth were assessed in the previous study conducted in Ethiopia. On the contrary, this study was based on primary data and included many risk factors which would be associated with preterm birth. In addition, most of the studies conducted in Ethiopia were cross-sectional and our study used case-control study design which is stronger than cross-sectional. Therefore, this study aimed to identify determinants of preterm birth in western part of Ethiopia. These informations are shortened in the abstract as follows: “There were lacking of study in western Ethiopia and most of those studies conducted in other parts of a country were based on card review with cross-sectional study design. The risk factors of preterm birth may vary from region to region within the same country due to variation in socioeconomic status and health care service coverage, and therefore this study aimed to identify determinants of preterm birth in western part of Ethiopia”.

B) Method: Line 37; the term “Data” is plural word. So, replace “was” with “were”.

Response: Replaced in this revised submission.

C) Conclusion: please propose certain recommendations based on your key findings.

Response: We agree with a reviewer that our previous conclusion lacks recommendation. Based on your suggestion, we gave recommendation in this revised submission as follows: “Antenatal care service providers should focus on mothers with pregnancy-induced hypertension, premature rupture of membrane and anemia during pregnancy, and refer to the senior experts for early management in order to reduce risk of preterm delivery. Antenatal care services such as counseling the mother on the benefit of dietary supplementation during pregnancy, antenatal care follow up and lengthening birth interval should be integrated into the existing health extension packages. Giving emphasis to these determinants with appropriate care during pregnancy is essential to reduce the occurrence of preterm birth. Increasing the awareness of contraceptive utilization and counseling to enhance birth spacing, antenatal care visits, folic acid and dietary supplementation during pregnancy should be given strict attention by healthcare providers. New and inclusive strategies such as establishment of comprehensive mobile clinic services should also be designed to reduce the burden of PTB among women living in the rural community. Lastly, we recommend future researchers to conduct longitudinal and community-based studies supplemented with qualitative methods”.

Introduction

A) I kindly recommend you to summarize the statement of the problem with 4 paragraphs: 1) the nature of the problem, 2) the known aspects of the problem (causes, magnitude risk factors, intervention (i.e., what is known?)), 3) the unknown aspects (any knowledge, intervention, methodology gaps (i.e., what is unknown?)), and 4) the expectation (s) from this study (i.e., so what?)

B) Please work out to improve the coherence.

Response: This is really very interesting comment and even teaching in our future research. Based on your recommendation, we tried to summarize our introduction as per your request in this revised manuscript.

C) I kindly request you for clarification: Line 55-57; it has to be clear in which regions or countries this classification works? For Example, in Ethiopia context, any termination of pregnancy prior to 28 completed weeks of gestational age is termed as "abortion". 

Response: We took this classification of preterm birth from a review of the global/international epidemiology of preterm births. We put this classification in the background of our manuscript just to show as PTB has sub-category based on gestational age. However, we agree with a reviewer that this classification makes confusion in differentiating abortion from preterm birth especially in Ethiopian context and therefore, we removed this sentence from this revised manuscript.

D) Please correct flaring errors. For example, line 58 states that “about 15 million infant born prematurely.” Herein, please change the infant to its plural form (i.e., infants). Ditto for line 86; change “literatures” to literature.

Response: We made these two changes in this revised submission.

E) Line 64-66; could you please provide information on the aggregate prevalence of preterm birth in Ethiopia.

Response: There is no systematic review conducted to shows the aggregate prevalence of preterm birth in Ethiopia. One systematic review and meta-analysis protocol on preterm birth was published, but its finding is not published yet. So, it is difficult to provide the aggregate prevalence of preterm birth in Ethiopia. However, we tried to provide the number of estimated preterm birth occurred each year in Ethiopia and other individual studies conducted in different parts of a country. 

Methods:

A) Line 123-125; generally further elaborations would be included regarding the case definition, inclusion and exclusion criteria. The cases and controls would be defined clearly, ‘’ not reliable ultrasonography’ has to be measurable. What measures have been made when gestational age disparity happened across records from LNMP and “early ultrasound”. Have you included post term births to controls? Have you included those preterm births which have been terminated deliberately? In general, ACOG definition would be used. 

Response: Cases (preterm births) were women who gave birth after 28 weeks (fetal viability) and before 37 completed weeks of gestation from the first day of the last normal menstrual period. Controls were women who gave birth at and after 37 weeks of gestation from the first day of the last normal menstrual period. 

All immediate postnatal women who gave birth at the selected hospitals of Wollega zones during the study period were included in the study. Women with unknown last menstrual period (LMP) or not reliable ultrasonography (not early taken at ≤20 completed weeks of gestation) and unable to communicate due to serious medical illness were excluded. Not reliable ultrasonography was considered if the U/S was not taken at appropriate time (at ≤20 completed weeks of gestation). 

We used LMP date and ultrasonography finding (if performed at ≤20 completed weeks of gestation) to estimate GA. If the LMP date and ultrasound date don’t correlate/disparity happened, defaulting to ultrasound for GA assessment is required and therefore we took ultrasound date. When the LMP was unknown, the ultrasound date was taken. If the women’s LMP is unknown and the ultrasound measurement was not taken at appropriate time (at ≤20 completed weeks of gestation), we excluded the mother from the study because we can’t assess GA.

Post term births are not preterm birth and therefore it was included under controls. Yes, deliberately or medically terminated pregnancy before 37 weeks of gestation were also included as preterm birth in this study. All these informations were included in this revised manuscript

B) Line 128-132; It is not clear why the authors took reference from Ghana for sample size calculation as a number of studies are here in Ethiopia (For instance; look at https://doi.org/10.1155/2020/1854073).

Response: The study conducted by Abebayehu Melesew Mekuriyaw et al. in Amhara Region Referral Hospitals (https://doi.org/10.1155/2020/1854073 is very interesting (case-control) and would be used as refence in calculating sample size. However, it was published (10 January 2020) after we fixed the sample size and conception of this study. Apart from this study, all of other studies conducted in Ethiopia were cross-sectional (except study in Tigray and Amhara region) and cannot be used as a reference in sample size calculation. We tried to use these two studies (Tigray and Amhara region) as a reference in sample size calculation, but they gave us relatively small sample size and therefore, we used the study conducted in African countries (Ghana) which gave us relatively higher sample size (565) for better reproducibility of the finding. 

Result:

A) Line 198-199; “Thirty-nine (22.67%) of cases and 77 (22.38%) of controls had no formal education” please add the article “the” in before “cases” ” controls”

Response: “The” was added before “cases” ” controls” in this revised manuscript.

B) Line 201-202; “Concerning residence, about two-third of the cases 116 (67.44%) and 214 (62.21%) of the controls were rural dwellers” would be corrected as one of the following:

1) Concerning residence, about 116 (67.44%) of the cases and 214 (62.21%) of the controls were rural dwellers or 2) Concerning residence, about two-third (67.44%) of the cases and three-fifth (62.21%) of the controls were rural dwellers.

Response: Thank you for your recommendation and we correct as per your request in this revised manuscript.

C) Line 208; “… majority of the study participants, 128 (74.42%) of the cases 253 (73.55%) of the controls …” needs English language editing.

Response: The sentence was edited in this revised manuscript as follows: “Regarding the use of family planning, 128 (74.42%) of the cases and 253 (73.55%) of the controls had used family planning before the current pregnancy”.

D) The regression table (i.e., “table 4”) poses major issues: �One category of occupation is “others”, what “others” stands for? List them by using foot note. �One category of “Marital status” is “widowed”. However, the number of cases in this category is only one and the corresponding 95% CI is too wide (i.e., 0.029, 1.809). In other words, about 1780 differences observed between the upper and lower limit which violates one of the assumptions of the binary logistic regression model. So, I kindly recommend you to merge some categories of the variables with few numbers of participants in the cases and / or controls. Thereafter, you need to undertake re-analysis provided that all variables which are entered to the model should fulfill the assumptions of the Binary logistic regression model. 

Response: Thank you very much for these important comments. We indicated what “Others” stands for using foot note in this revised manuscript. Some of the categories of our variables have small observation in the outcome variable and their confidence intervals are wide. For example: mothers age (category ‘>44’ years) and marital status (category ‘widowed’) have small observations and wide CI. We didn’t account this problem during the analysis of our data. Therefore, based on your comments, we merged some categories of the variables with few numbers of participants in the cases and / or controls and undertake re-analysis to fulfill the assumptions of the Binary logistic regression. Marital status which was associated with PTB at Bivariable analysis in the previous analysis is now not significant, and mothers age which was previously not associated with PTB at bivariate analysis is now significant. However, no difference was seen on the variables which were associated with PTB at multivariable analysis (No variable which previously significant is non-significant in this new analysis and vise versa). But there are some differences in results of cross-tabulation, AOR, and P value between the former and this revised manuscript

About 50 cases and 87 controls were in the age group of 15-24 which implies that minors have been included in the study. If so, how have you addressed the ethical issues? In this occasion, let me raise one another important concern; why have you failed in incorporating a sub-section of “Ethical approval”?

Response: Sorry for not including ethics and participant consent in the previous version of our manuscript. We included these informations in this revised version of our manuscript as follows: 

“Ethics approval and consent to participate: The study was approved by the institutional review boards of Wollega University ethical review board with approval ID: HIS/213/20. Permission letter was also obtained from each hospital administrative office. All participants of the study were provided written consent, clearly stating the objectives of the study and their right to refuse. Then, written informed consent was obtained from the study participants. For minors, informed consent was received from their parents or legal guardians. To ensure confidentiality, names or identifying information was not indicated on the questionnaires. Mothers were interviewed in private rooms to ensure their privacy. The filled questionnaires were carefully handled ensuring confidentiality and was kept under secured custody of the corresponding author”.

The sum of each category of variables under the cases has to be 100%. Ditto for controls. However, that was not true in your study. This implies that you have considered the “row” percentages in the “crosstab” that is recommended for cross sectional study design. For case – control study design, however, “column” percentages should be reported over “row” percentages. Therefore, you need to address this concern while you perform re-analysis as per the above recommendation.

Response: Thank you again for this crucial comment and we performed as per your request in this revised manuscript.

Discussion:

Your way of discussion is interesting. However, still it needs revision for language:

A) Tense or spelling errors: For instance; line 274; change “remained” to be “remains”, line 309; edit “Gonder” to be “Gondar”

Response: Thank you for this important correction and we made it in this revised revision.

B) Certain phrases have been employed frequently. For example “the odds of developing preterm birth...”, “…due to the fact that…”

Response: We re-wrote these and other phrases to prevent the frequent use of these phrases.

C) Be consistent in using the abbreviations Vs the extended forms. For example; line 302; you have utilized “Pregnancy induced hypertension”, whereas at line 303; you have used its abbreviation for “PIH”, again at line 305, 307, 311…; you have employed its extended form “Pregnancy induced hypertension”???Moreover, the abbreviation “PIH “has not been listed under the “Abbreviations” section. On the contrary, the abbreviation “PIHTN” has been found under the list although it has not been used at the main document at all. I kindly recommend you to put the extended form of each abbreviation followed by its abbreviated form in bracket at its first appearance. Then you can use the abbreviated form alone throughout the document and don’t forget mentioning it at the lists of “abbreviations “section.

Response: Thank you for this important comment and we put as your request throughout the document. 

D) The limitations of your study as well as the efforts made to overcome those limitations would be stated.

Response: We forgot including limitation of the study in the previous submission. We included limitation of this study in this revised submission as follows: “There may be recall bias from mothers due to the nature of some question which dealt with past informations. To reduce this recall bias, information gathered from the mothers through the interview were crosschecked from their antenatal records. Due to the sensitive nature of some questions and face-to-face techniques of data collection, there is a possibility of falsified reporting (social desirability bias) among mothers. We made an effort to minimize this by assuring mothers for the confidentiality of their informations. Another limitation of the study is being institution-based quantitative study. It would be better if community-based and qualitative approach study was triangulated with quantitative part to investigate further factors on preterm birth. On the other hand, the strength of this study is we used strong design with large sample size with 1:2 ratio of cases to controls”. 

References:

A) Reference 6(line 374-376) & 28 (line 430-432) are similar.

Response: Thank you for this comment. We removed reference 28 from the list in this revised submission and replaced with other. 

B) You would use certain recently published articles. For example “Determinants of Preterm Birth among Women Who Gave Birth in Amhara Region Referral Hospitals, Northern Ethiopia, 2018: Institutional Based Case Control Study”, which is available at https://doi.org/10.1155/2020/1854073 , has not been cited in your manuscript.

Response: We thorough read this article and we found it very interesting. It was recently published (10 January 2020) and we didn’t get this article when finalizing our research. So, based on your request, we now included and used as a reference in this revised manuscript. We used it as reference number “38” and cited as follows: “Mekuriyaw AM, Mihret MS, Yismaw AE. Determinants of Preterm Birth among Women Who Gave Birth in Amhara Region Referral Hospitals, Northern Ethiopia, 2018: Institutional Based Case Control Study. International Journal of Pediatrics; Volume 2020, Article ID 1854073, 8 pages https://doi.org/10.1155/2020/1854073”

 Thank you very much for your crucial comments!

---

## [Decision Letter · Decision Letter 1]

8 Dec 2020

PONE-D-20-17494R1

Determinants of preterm birth among women delivered in public hospitals of Western Ethiopia, 2020: Unmatched case-control study

PLOS ONE

Dear Dr. Abadiga,

Thank you for submitting your manuscript to PLOS ONE. After careful consideration, we feel that it has merit but does not fully meet PLOS ONE’s publication criteria as it currently stands. Therefore, we invite you to submit a revised version of the manuscript that addresses the points raised during the review process.

We encourage you to let the manuscript be checked for proof-reading by a native English speaker.

We look forward to receiving your revised manuscript.

Kind regards,

Florian Fischer

Academic Editor

PLOS ONE

Reviewers' comments:

Reviewer's Responses to Questions

**Comments to the Author**

1. If the authors have adequately addressed your comments raised in a previous round of review and you feel that this manuscript is now acceptable for publication, you may indicate that here to bypass the “Comments to the Author” section, enter your conflict of interest statement in the “Confidential to Editor” section, and submit your "Accept" recommendation.

Reviewer #2: All comments have been addressed

2. Is the manuscript technically sound, and do the data support the conclusions?

Reviewer #2: Yes

3. Has the statistical analysis been performed appropriately and rigorously? 

Reviewer #2: Yes

4. Have the authors made all data underlying the findings in their manuscript fully available?

Reviewer #2: Yes

5. Is the manuscript presented in an intelligible fashion and written in standard English?

Reviewer #2: Yes

6. Review Comments to the Author

Reviewer #2: Manuscript number: PONE-D-20-17494R1

This paper has been greatly improved and almost all my concerns have been addressed. However, the manuscript has still certain glaring errors. For example, line 25-26 “…1 million death…and 60% of this death occur in ….” has to be corrected as …1 million deaths… and 60% of these deaths occur in…

Therefore, the manuscript needs to be revised for language thoroughly.

7. PLOS authors have the option to publish the peer review history of their article (what does this mean?). If published, this will include your full peer review and any attached files.

Reviewer #2: No

---

## [Author Response · Author response to Decision Letter 1]

13 Dec 2020

Date: 13/12/2020

To PLOS ONE

Dear Academic editor, 

Subject: Submission of revised Manuscript “Determinants of preterm birth among women delivered in public hospitals of Western Ethiopia, 2020: Unmatched case-control study”. Thank you very much for reviewing our manuscript. We greatly appreciate the editor and reviewers for their constructive comments and suggestions. We have carried out the revisions that the editor and reviewers requested and revised the manuscript accordingly. We hope the revised version is now suitable for publication and look forward to hearing from you in due course. 

Sincerely,

Muktar Abadiga 

Corresponding author 

Wollega University, Ethiopia

Point by point response

 Editor comments:

We encourage you to let the manuscript be checked for proof-reading by a native English speaker.

Response: Thank you very much for your suggestion to improve our manuscript. Based on your recommendation, we contacted native English speaker to correct the grammar, and we hope this version of our manuscript is now suitable for publication.

Reviewer comments:

Reviewer #2: Manuscript number: PONE-D-20-17494R1

This paper has been greatly improved and almost all my concerns have been addressed. However, the manuscript has still certain glaring errors. For example, line 25-26 “…1 million death…and 60% of this death occur in ….” has to be corrected as …1 million deaths… and 60% of these deaths occur in… Therefore, the manuscript needs to be revised for language thoroughly.

Response: Thank you very much once again for your great contribution in the improvements of this manuscript. The previous version of our manuscript has some grammatical and editorial problems. In this revised submission, all authors intensively reviewed the document and corrected grammatical and editorial problems. We also used free online grammar correction to solve this problem and contacted native English speaker to correct the grammar, and we hope this version of our manuscript is now suitable for publication.

 Thank you very much

---

## [Editor Report · Decision Letter 2]

17 Dec 2020

Determinants of preterm birth among women delivered in public hospitals of Western Ethiopia, 2020: Unmatched case-control study

PONE-D-20-17494R2

Dear Dr. Abadiga,

We’re pleased to inform you that your manuscript has been judged scientifically suitable for publication and will be formally accepted for publication once it meets all outstanding technical requirements.

Kind regards,

Florian Fischer

Academic Editor

PLOS ONE
---

## [Editor Report · Acceptance letter]

14 Jan 2021

PONE-D-20-17494R2 

Determinants of preterm birth among women delivered in public hospitals of Western Ethiopia, 2020: Unmatched case-control study 

Dear Dr. Abadiga:

I'm pleased to inform you that your manuscript has been deemed suitable for publication in PLOS ONE. Congratulations! Your manuscript is now with our production department. 

Kind regards, 

on behalf of

Dr. Florian Fischer 

Academic Editor

PLOS ONE